# Impact of Thermal and High-Pressure Treatments on the Microbiological Quality and *In Vitro* Digestibility of Black Soldier Fly (*Hermetia illucens*) Larvae

**DOI:** 10.3390/ani10040682

**Published:** 2020-04-14

**Authors:** Mairead Campbell, Jordi Ortuño, Alexandros Ch. Stratakos, Mark Linton, Nicolae Corcionivoschi, Tara Elliott, Anastasios Koidis, Katerina Theodoridou

**Affiliations:** 1Institute for Global Food Security, Queen’s University Belfast, Belfast BT9 5AJ, Northern Ireland, UK; mcampbell105@qub.ac.uk (M.C.); j.ortunocasanova@qub.ac.uk (J.O.); telliott08@qub.ac.uk (T.E.); t.koidis@qub.ac.uk (A.K.); 2Faculty of Health and Applied Sciences, Center for Research in Biosciences, University of the West of England, Bristol BS16 1QY, UK; Alexandros.Stratakos@uwe.ac.uk; 3Agri-Food & Bioscience Institute, Belfast BT9 5PX, Northern Ireland, UK; mark.linton@afbini.gov.uk (M.L.); nicolae.corcionivoschi@afbini.gov.uk (N.C.)

**Keywords:** insects, animal feed, high-pressure processing, microbial safety, *in vitro* digestibility, black soldier fly

## Abstract

**Simple Summary:**

While facing climate change and natural resource scarcity, ensuring sufficient, nutritious, safe, and affordable protein sources to a fast-growing feed demand becomes increasingly challenging. The emerging insect sector has the potential to improve the circularity of the agri-food chain thanks to their ability to upcycle industrial organic wastes into valuable biomass that can be included as a feed ingredient for livestock. The black soldier fly is considered one of the most promising insect species for its large-scale production due to its ability to be reared in a wide variety of organic substrates. However, more information is required regarding the suitability of agri-food by-products and processing techniques to ensure the quality of the final insect-derived products for large-scale production. The present study showed that breweries’ by-products are a suitable source of substrate for the development of black soldier fly larvae as an ingredient for both ruminant and non-ruminant livestock feed. High-pressure processing showed no clear improvement in terms of decontamination capacity and digestibility in comparison to heating treatment, resulting in a less cost-effective process for large-scale production of black soldier fly larvae.

**Abstract:**

Black soldier fly larvae (BSFL) are gaining importance in animal feeding due to their ability to upcycle low-value agroindustry by-products into high-protein biomass. The present study evaluated the nutritional composition of BSFL reared on brewer’s by-product (BBP) and the impact of thermal (90 °C for 10/15 min) and high-pressure processing (HPP; 400/600MPa for 1.5/10 min) treatments on the microbial levels and *in vitro* digestibility in both ruminant and monogastric models. BBP-reared BSFL contained a high level of protein, amino acids, lauric acid, and calcium, and high counts of total viable counts (TVC; 7.97), Enterobacteriaceae (7.65), lactic acid bacteria (LAB; 6.50), and yeasts and moulds (YM; 5.07). Thermal processing was more effective (*p* < 0.05) than any of the HPP treatments in reducing TVC. Both temperature of 90 °C and pressure of 600 MPa reduced the levels of Enterobacteriaceae, LAB, and YM below the detection limit. In contrast, the application of the 400 MPa showed a reduced inactivation (*p* < 0.05) potential. Heat-treated samples did not result in any significant changes (*p* > 0.05) on any of the *in vitro* digestibility models, whereas HPP showed increased and decreased ruminal and monogastric digestibility, respectively. HPP did not seem to be a suitable, cost-effective method as an alternative to heat-processing for the large-scale treatment of BSFL.

## 1. Introduction

Sustainable protein sources for livestock diets are urgently required to address the future challenges of feeding an estimated global human population of 9 billion by 2050 [1]. In recent years, insects are being considered as one of the most promising alternatives to common protein feed ingredients, such as soybean and fish meals. After the approval of the use of insect-derived products in fish feed [2], it is expected that the European Union (EU) will extend the authorization for other food-producing animals during the following years. Black Soldier fly (BSF; *Hermetia illucens*) is one of the main species considered for large-scale insect farming given their ability to efficiently upcycle organic waste into a high-value protein source, thus increasing both the productivity and the efficiency of the food chain [3]. 

Industry organic by-products (i.e., breweries, ethanol, oil industries) offer great possibilities as a substrate for the large-scale production of BSF, as they are supplied continuously, in large quantities, and with a consistent nutritional composition. Nevertheless, the nutritional content of the agri-food residues does not always fit the requirements of BSF for the production of protein-rich biomass suitable as feed ingredients for pigs, chickens, and fish [4]. Moreover, insect-derived product digestibility depends not only on the insect species and rearing substrate, but also on the processing method and conditions (time, temperature) applied [5,6,7,8,9]. Within this context, there is a lack of research regarding cost-effective rearing methods and post-harvest processing technologies for the mass production of insects on an industrial scale, in order to ensure both the safety of the product and the preservation of its nutritional value and digestibility [10]. 

Heat treatment represents an effective step in reducing the initially high counts of microorganisms found in fresh insects, as well as in improving the digestibility and bioavailability of some nutrients in the digestive tract [11]. However, the nutritive value of insects can also be negatively affected, as thermal processing (TP) may induce proteolysis, lipolysis, lipid oxidation, and loss of vitamins [12]. High-pressure processing (HPP) is a volumetric, non-thermal, pasteurizing method that uses pressures of between 150 to 600 MPa. HPP results in an effective reduction of microbial counts with minimal effect on nutritional content, although it may also induce structural changes in food systems, especially when pressures higher than 400 MPa are used [13]. Therefore, HPP and TP may be useful in reducing the microbial load of BSF larvae, although they may differently affect monogastric and ruminant digestibility. The aims of this study were (i) the evaluation of the nutritive value of the BSF larvae reared on brewer’s by product (BBP) as a potential feed ingredient, and (ii) the investigation of the effects of TP and HPP treatments on black soldier fly larvae (BSFL) microbial levels and *in vitro* dry matter digestibility employing ruminant and monogastric models. 

## 2. Materials and Methods 

### 2.1. Preparation of Larvae

Live BSF larvae (BSFL) reared on dried BBP were supplied by Hexafly (Co. Meath, Ireland). After sieving to remove the frass, the larvae were fasted for 24 h at room temperature to empty the gut content, killed by freezing at −18 °C for 1 hour, and vacuum-packed before processing. All processing methods were applied in triplicate to the same batch of insects. 

### 2.2. Chemical Composition

Dry matter (DM; 930.15), ash (942.05), and ether extract (EE; 954.02) were measured according to Association of Official Analytical Chemists (AOAC) official methods [14]. Neutral detergent fibre (aNDF) and acid detergent fibre (ADF) were analysed by consecutive detergent analysis [15], using the ANKOM 220 Fibre Analyzer (ANKOM Technology Corporation, Macedon, NY, USA) with sodium sulphite and heat-stable α-amylase; the results were expressed inclusive of ash. Lignin was analyzed by the direct sulphuric acid method, according to Robertson and Van Soest [16]. Nitrogen content was analysed by the Dumas method using Leco Protein/N Analyser (FP-528, Leco Corp., St Joseph, MI, USA), and crude protein (CP) was calculated using Nx6.25. The mineral composition was measured using an energy dispersive X-ray fluorescence spectrometer (NEX CG, Rigaku Instruments); bovine muscle was used as certified reference material (percentage of recovery was considered acceptable with a variability of ±20%). Chemical and mineral analyses were performed on freeze-dried larvae in triplicate, except for amino acid (AA) and fatty acid (FA) composition (single analyses). AA composition was determined by HPLC performed on oxidized and hydrolysed samples, except for tryptophan, following the procedure in [17]. FA profile was assessed by gas chromatography (GC) [18].

### 2.3. Thermal Processing (TP)

Frozen larvae were thawed and subjected to two thermal processes: 90 °C for 10 (TP10) and 15 (TP15) min. The inner core temperature of the bags was monitored by a Thermocouple Datalogger AZ9881 K (ETI Ltd., West Sussex, United Kingdom). The larvae were placed in a water bath (SBB28, Grant Instruments Lt., Cambridge, United Kingdom) at 100 °C to allow a rapid temperature increase up to 90 °C. Once a temperature of 90 °C was achieved, the sample was transferred to a second water bath set to 90 °C and held for 10 or 15 min, and finally cooled to ambient temperature in an ice bath.

### 2.4. High-Pressure Processing (HPP) 

Pressurization treatments were conducted at ambient temperature (18 °C) using a commercial scale high-pressure press (Quintus 35L, Quintus Technologies AB, Västerås, Sweden) with a pressure vessel of 35 L volume. Come-up time to achieve maximum pressure was approximately 25 s per 100 MPa, and the pressure release time was approximately 10–17 s on the basis of the pressure settings. The pressurizing medium was potable water, and the temperature increase due to adiabatic heating was around 2.2 °C per 100 MPa. The larvae were subjected to four treatments: (a) 400 MPa for 1.5 min (P400/1.5), (b) 400 MPa for 10 min (P400/10), (c) 600 MPa for 1.5 min (P600/1.5), and (d) 600 MPa for 10 min (P600/10).

### 2.5. Microbiological Analyses

After treatment, samples were opened aseptically, and the contents were transferred to a sterile stomacher bag (Interscience, St. Nom La Breteche, France). A 10^−1^ dilution of the sample was prepared in maximum recovery diluent (Oxoid, CM733, Basingstoke, United Kingdom). The dilution was homogenized for 60 s at high speed in a Seward stomacher (Lab blender 400, Bury St. Edmunds, United Kingdom). Further 10-fold dilutions were prepared in 9 mL Maximum Recovery Diluent (MRD). Total viable counts (TVC) were enumerated by spread-plating onto tryptone soya agar with 0.6% yeast extract plates (TSAYE; Oxoid, Basingstoke, United Kingdom), with aerobic incubation at 30 °C for 48 h. Lactic acid bacteria (LAB) were enumerated on de Man Rogosa and Sharp agar (Oxoid, Basingstoke, United Kingdom) by pour-plating with overlay and incubating aerobically at 30 °C for 72 h. Enterobacteriaceae were enumerated using Violet Red Bile Glucose Agar (Oxoid, Basingstoke, United Kingdom) by pour-plating with overlay and incubating aerobically at 37 °C for 72 h. Yeasts and moulds (YM) were enumerated on Dextrose Rose-Bengal Chloramphenicol agar (Oxoid, Basingstoke, United Kingdom) with incubation at 25 °C for 72 and 120 h. Each sample was plated in duplicate, and the results were expressed as log colony-forming units (CFU)/g.

### 2.6. In Vitro Dry Matter Digestibility 

For the apparent total tract dry matter monogastric digestibility (ATTD), a three-step method was applied simulating the gastric, small intestine, and large intestine digestion [19]. A sample (0.5 ± 0.1 g) was weighed into a 100 mL glass flask with a screw cap. Phosphate buffer (25 mL, 0.1 N, and pH 6.8) and 10 mL of 0.2 N HCl solution were added to the flask, and the pH was adjusted to 2 using 1 N HCl or 1 N NaOH. A total of 1 mL (25 mg/mL) of freshly prepared pepsin (P-7000, Sigma-Aldrich, St. Louis, MO, USA) and 0.5 mL of chloramphenicol (0.5 g/100 mL of ethanol) were then added, and the flasks were incubated at 39 °C for 2 h with magnetic agitation. After the incubation, solutions of 10 mL of 0.2 N phosphate buffer (pH 6.8) and 5 mL of 0.6 N NaOH were added to the flask, and the pH of the solution was adjusted to 6.8 with a 1 N HCl or 1 N NaOH solution. After that, 2.5 mL (40mg/mL) of freshly prepared pancreatin (P-1750, Sigma-Aldrich) solution was added to the flask and incubated in a water bath at 39 °C for 4 h. After the second incubation, 10 mL of a 0.2 M EDTA solution was added to the flask, and the pH was adjusted to 4.8 with 30% acetic acid solution. Then, 0.5 mL of Viscozyme (multienzyme complex from *Aspergillus aculeatus* containing cellulase,) was added, and the flask incubated again at 39 °C for 18 h. After the last incubation phase, 5 mL of 20% sulfosalicylic acid was added into the flasks and left for 30 min at room temperature. Then, flask contents were filtered using pre-dried (80 °C) Whatman no. 54 filter papers (Whatman Inc., Florham Park, NJ). The *in vitro* residues collected in filter papers were also dried overnight at 80 °C. ATTD was calculated by deducting the residue DM from the sample DM, followed by division by the sample DM. The *in vitro* ruminal true dry matter digestibility (IVTDMD) was determined on the basis of the instructions of the DaisyII incubator (ANKOM, NY, USA) [20]. Rumen contents were collected 3 hours after morning feeding from two Holstein cows fitted with a ruminal cannula, at Agri-Food & Bioscience Institute (AFBI) Hillsborough, in Co. Down, Northern Ireland (United Kingdom). The digesta from both cows was transported to the laboratory, homogenized, filtered through four layers of cheesecloth, and purged with CO_2_. The rumen fluid was then poured into the jars of the DaisyII incubator which were previously filled with warmed (39 °C) buffer solution (v/v as 1:5; pH 6.8). At the end of the 48 h incubation, the bags were rinsed four times with distilled water. For *in vitro* true digestibility determination, the bags were placed in an ANKOM fibre analyser and boiled in neutral detergent solution for 1 hour. The bags were removed, soaked twice in acetone for 5 min, and dried at 105 °C for 3 h. The IVTDMD was calculated as the difference between DM incubated and the residue after neutral detergent (ND) treatment. Both digestibility assays were replicated three times for each treatment replicate for a total of nine digestions.

### 2.7. Statistical Analysis

Data were analysed using JMP 14.0 (SAS Institute, Cary, NC, USA). One-way analysis of variance (ANOVA) was used to compare differences in the treatment means. When the effect was significant (*p* < 0.05), post hoc comparisons were carried out using Tukey’s HSD multiple comparisons to calculate the least significant differences among the means. 

## 3. Results

### 3.1. Nutritional Composition

Table 1 shows the nutritional composition of whole untreated BSFL. The content of CP (49.1% DM), EE (36.6% DM), and ash (5.45% DM) fell within the range (46.3–54.2%, 29.9–38.6%, 4.8–11.6%, respectively) of previous values of BSFL reared on a similar substrate [21,22,23]. BSFL showed a high level of essential AA (Table 2), including lysine (2.60 g/100 g DM), methionine (0.84 g/100 g DM), and tryptophan (0.60 g/100 g DM), above those reported for full-fat soybeans with similar protein content [24]. In comparison with previous studies on BSF, the AA profile of BBP-reared larvae agreed to those found when diverse substrates were employed [25,26]. BSF seems to accumulate glutamic and aspartic acids as the main AA, representing values between 21–28% of total AA [24,25,26], such as in our study (23.1%). Likewise, leucine, lysine, and valine were the most prevalent among the essential AA, with values consistent with BSF reared on similar and diverse substrates [25,26,27]. Overall, our results support the idea that the BSF AA profile has low variations independently of the substrate employed [26]. Indeed, Liland et al. [27] found only small variations on AA composition when BSFL diet changed from 100% wheat to 100% seaweed. 

In contrast, FA profile (Table 3) and mineral content (Table 1) tend to be more dependent on the substrate [27]. In the present study, the overall content of saturated (SFA) (64.5%), monounsaturated (MUFA) (12.3%), and polyunsaturated fatty acids (PUFA) (19.6%), as well as the major individual FA (C12:0 > C18:2n6 ≥ C16:0 > C18:1), were similar to previous studies using spent grains as substrates [21,23]. In comparison with other substrates, the content of FA groups and individual FA is highly variable and seems to be influenced not only by the fat content but also by the carbohydrate fraction, as insects can convert carbohydrates into lipids [21]. Substrates rich in non-structural carbohydrates, such as fruit wastes, tend to accumulate higher SFA content than agro-industrial by-products similar to BBP [21,26]. However, in all cases, lauric acid seems to be the most prevalent SFA. The high content of C12:0 might be beneficial for animal production, as it is absorbed and metabolized faster than long-chain FA, implying an efficient energy source, especially for young animals, as well as for the antimicrobial effect shown against Gram-positive pathogenic gastrointestinal bacteria, such as *Streptococcus suis* [28]. For this reason, the inclusion of BSF as a feed ingredient is highlighted as a potential alternative to reduce the use of antimicrobials in piglets’ diets. In contrast, the low PUFA n3 content in BSF might be detrimental for the nutritional quality of animal-derived food products when used as animal feed [5]. Nevertheless, the FA profile may also be modulated by the substrate, thus increasing the versatility of BSF as raw feed material [29].

BSFL showed a high content of relevant macro (Ca, P, K, Mg) and micro minerals (Mn, Fe, Zn), with a Ca/P ratio of 4.2. Ca level, the major mineral of BSFL, is dependent on the substrate and tends to increase in the pre-pupae and pupae stage [30]. The calcium content of BSFL found in the present study was higher (18.5 g/kg DM) than in previous studies using a similar substrate (5.36 g/kg [31]; 1.70 g/kg [22]) but lower than common values observed in BSFL (50–86 g/kg [3]). The bioaccumulation of heavy metals in farmed insects has been highlighted as a potential risk for animal feed [32]. In the present study, cadmium and arsenic were not found, and lead (3.6 mg/kg DM) was below the maximum limit of 10 mg/kg (88% DM) permitted by the EU [33]. The content of chitin in BSFL can be estimated by detergent fibre results (ADF−ADL = 6.60 g/100 g DM; [34]). Chitin has been associated with prebiotic and antimicrobial properties, although it may also be considered as an antinutritional factor when high content is included in the diet [5]. Overall, BSFL reared on BBP showed a suitable CP, AA, and mineral composition for both ruminant and monogastric animals. However, to enhance the applicability of BSFL as animal feeding as replacement of soy and fishmeal, the fat content might be reduced, which would increase its CP and AA content. However, some precaution should be taken as high levels of ash and chitin may be detrimental to animal production [3]. 

### 3.2. Microbial Analysis

Table 4 shows the effect of the different combinations of treatment and time on the microbial quality of the whole BSFL. Microbial counts were high in the fresh samples, with TVC and Enterobacteriaceae counts higher than 7 log CFU/g, and LAB and YM levels reaching 6.50 and 5.07 log CFU/g, respectively. Previous studies already showed similar counts for all the microbial species analysed in the whole BSFL after freeze-killing [12,35,36]. The total bacterial population found in insects include both surface and intestinal bacteria. Therefore, apart from the processing, the original contamination of the rearing substrate will directly influence the total level and type of microflora present. All the treatments applied reduced (*p* < 0.05) the BSFL microbial load of all the microorganisms considered, except for the mildest HPP treatment (P400/1.5), which showed no reduction (*p >* 0.05) of the TVC and LAB in comparison with C samples. Both TP, regardless of the time applied (*p* > 0.05), achieved the highest inactivation effect, with a maximum of 2.45 log CFU/g on TVC counts, and values below the detection limit (2 log CFU/g) for Enterobacteriaceae, LAB, and YM. Therefore, TP10 was effective enough for a reduction higher than 5.65, 4.50, and 3.07 log CFU/g for Enterobacteria, LAB, and YM, respectively. Regarding the HPP treatments, the effects were dependent on pressure level and exposure time. P400/10 (1.44 log CFU/g) and P600/10 (1.30 log CFU/g) achieved higher reduction (*p* < 0.05) for TVC compared to the same pressure applied for 1.5 min. HPP treatment for 1.5 min showed only a slight reduction in the TVC counts, and only the counts for samples treated with P600 were lower (0.59 log; *p* < 0.05) than the C samples. In contrast, for LAB, Enterobacteriaceae, and YM, both P600 treatments showed a significant (*p* < 0.05) reduction of microbial counts, similar (*p* > 0.05) to the one achieved by TP. P400/10 min showed similar potential in reducing YM levels but was less effective compared to the heat treatment and P600 for reducing Enterobacteria and LAB. These results are in line with those of Kashiri et al. [36], who also found a limited capacity of HPP at 250 and 400 MPa to reduce the TVC levels of fresh BSFL, with a maximum reduction of 0.35 log CFU/g after 7 min at 400 MPa, and a higher effect on YM, with no survivors when 400 MPa was applied for longer than 2.5 min. Rumpold et al. [11] employed similar heating and HPP treatments for assessing the decontamination potential on *Tenebrio molitor* larvae. Similarly, thermal treatments at 90 °C achieved better results for the inactivation of the overall microbial counts. Both heating and HPP are volumetric treatments able to reduce both the surface and internal gut microbiota in insects. However, insects have been identified as potential vectors of spore-forming bacteria, which could not be eliminated by pasteurization treatments [32]. The effect of HPP treatments on spore-forming bacteria is dependent on time and temperature applied, and for spore inactivation, a combination of elevated pressures (>400 MPa) and elevated temperatures (>60 °C) are necessary [37]. Therefore, none of the treatments applied in the present study, which was conducted at ambient temperature, could reduce spore-forming bacteria in BSFL, which would explain the limited effect on TVC. According to the results, HPP would not be viable for reduction of TVC on the scale required, and it is unlikely to be cost-effective compared with conventional heat treatment if there are no significant advantages in terms of digestibility or other nutritional benefits.

### 3.3. In Vitro Digestibility 

Figure 1 shows the results of the both *in vitro* monogastric ATTD and ruminant IVTDMD of whole BSFL. In vitro digestibility techniques using enzymes and lengths of incubations that mimic in vivo digestion of monogastric animals have been commonly used to assess the ATTD of different feedstuffs with good accuracy and repeatability [38]. Previous studies have shown that the effect of different heat-processing techniques on insect digestibility in similar monogastric models is influenced by the conditions applied and the species considered [6,8]. In the present study, ATTD values of the untreated insects (0.84) were similar (0.81) to those found previously for BSFL by Bosch et al. [25]. After TP, the ATTD values were unaffected (*p* > 0.05), whereas HPP showed a decreasing trend, which was more pronounced after application of 400 MPa (*p* < 0.05) (0.79) compared to 600 MPa (0.81). Exposure time did not affect (*p* > 0.05) any treatment. Poelaert et al. [39] observed a reduction in the ATTD for the house cricket and yellow mealworm after heating treatments using similar *in vitro* models. Nevertheless, the treatments applied in that study used stronger conditions (oven-roasted at 150/200 °C and autoclaving) than the ones employed in the current study, which should have prevented relevant physicochemical modifications (oxidation, aggregation, Maillard reaction) of the larvae constituents. 

The ruminal IVTDMD of untreated BSFL was high (0.88), indicating its potential as a digestible component in the ruminant diet. In contrast, Jayanegara et al. [30] found a lower IVTDMD in BSFL (<0.56) and other insect species (Jamaican field cricket—0.64, and mealworm—0.60). As in the case of the monogastric ATTD, TP did not affect (*p* > 0.05) the IVTDMD. However, different trends were observed after the application of HPP, as IVTDMD seemed to increase (*p* < 0.05) after the application of both 400 and 600 MPa, to a maximum of 5%. During HPP, different processes can occur at the same time, including enzyme-catalyzed conversion processes, chemical reactions, and modification of biopolymers, depending on the product and parameters applied [13]. However, more research should be done to investigate the nutritional composition and digestibility of individual compounds (CP, AA, NDF) before we can conclusively assess the potential effect of HPP on BSF composition and the monogastric and ruminant digestibility.

## 4. Conclusions

BSFL reared on BBP showed a suitable nutritional composition to be considered as a promising animal feed ingredient as replacement of conventional protein sources. TP resulted in higher inactivation values than HPP for TVC, and a pressure of 600 MPa was needed to achieve similar reductions against Enterobacteria, LAB, and YM. TP did not affect digestibility either in monogastric or ruminant models, whereas HPP increased and reduced the ruminant and monogastric digestibility, respectively. The effect of HPP on the nutritional components (structural carbohydrates, protein, and amino acid content) and combination of HPP and temperature on spore-forming bacteria would be needed to assess its true potential as a cost-effective method to improve the use of BSFL as a feed ingredient. 

## Figures and Tables

**Figure 1 animals-10-00682-f001:**
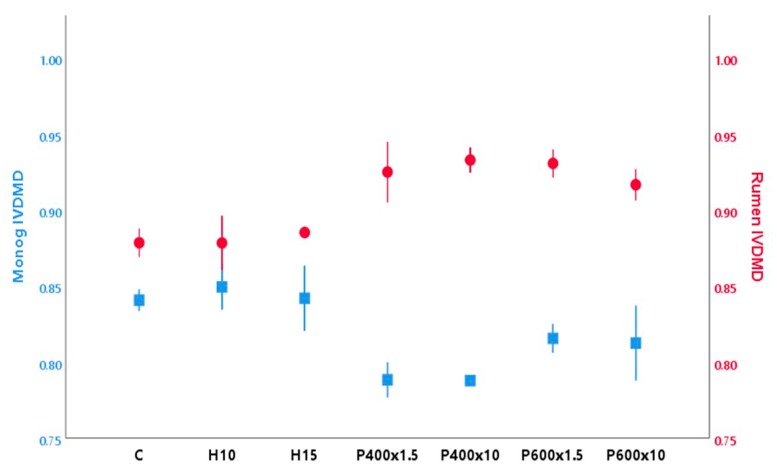
Mean values (± standard deviation) of BSFL *in vitro* true rumen (●) and apparent monogastric total tract (■) dry matter digestibility after the application of different treatments.

**Table 1 animals-10-00682-t001:** Nutritional (g/100 g dry matter (DM), unless stated) and mineral composition (g/kg DM) of untreated black soldier fly larvae (BSFL) reared in brewer’s by-product (mean ± standard deviation).

**DM (g/100g)**	**CP**	**EE**	**Ash**	**aNDF**	**ADF**	**ADL**
34.1 ± 0.19	49.1 ± 0.51	36.6 ± 0.69	5.45 ± 0.21	13.0 ± 1.04	7.92 ± 0.48	1.45 ± 0.12
**Ca**	**K**	**Mg**	**P**	**Fe**	**Mn**	**Zn**
18.5 ± 0.38	6.91 ± 0.19	2.87 ± 0.11	4.40 ± 0.19	0.15 ± 0.04	0.12 ± 0.03	0.11 ± 0.04

DM = dry matter; CP = crude protein (calculated as N × 4.73); EE = ether extract; aNDF = neutral detergent fibre; ADF = acid detergent fibre; ADL; lignin.

**Table 2 animals-10-00682-t002:** Amino acid profile (g/100 g DM) of BSFL reared in brewer’s by-product.

Alanine	2.44	Lysine	2.60
Arginine	2.15	Methionine	0.84
Aspartic acid	3.95	Phenylalanine	1.78
Cystine	0.36	Proline	2.08
Glutamic acid	4.75	Serine	1.66
Glycine	2.25	Threonine	1.62
Histidine	1.33	Tryptophan	0.60
Isoleucine	1.75	Tyrosine	2.26
Leucine	2.78	Valine	2.41

**Table 3 animals-10-00682-t003:** Fatty acid profile (g/100 g total fatty acids - TFA) of BSF larvae fed with brewer’s by-product.

C10:0	0.92	C16:1	2.29	SFA	64.5
C12:0	36.0	C18:0	2.28	MUFA	12.3
C14:0	7.83	C18:1	9.38	PUFA	19.6
C14:1	0.15	C18:2n6	18.1		
C15:0	0.21	C18:3n3	1.48		
C15:1	0.10	C20:0	0.11		
C16:0	16.8	C20:1	0.19		

Fatty acids with values < 0.05 g/100 g TFA were omitted.

**Table 4 animals-10-00682-t004:** Effect of different treatments on the microbial levels (log colony-forming units (CFU)/g) of black soldier fly larvae

Treatment	Control	TP		HPP				SEM	*p*
Pressure (MPa)				400		600			
Time (min)		10	15	1.5	10	1.5	10		
TVC	7.97 ^a^	5.52 ^e^	5.63 ^e^	7.65 ^a,b^	6.53 ^d^	7.28 ^b,c^	6.67 ^c,d^	0.201	<0.001
ENB	7.65 ^a^	<2 ^d^	<2 ^d^	6.12 ^b^	3.32 ^c^	2.09 ^d^	<2 ^d^	0.485	<0.001
LAB	6.50 ^a^	<2 ^c^	<2 ^c^	6.60 ^a^	4.73 ^b^	<2 ^c^	<2 ^c^	0.476	<0.001
YM	5.07 ^a^	<2 ^c^	<2 ^c^	3.36 ^b^	<2 ^c^	<2 ^c^	<2 ^c^	0.265	<0.001

TP = thermal processing; HPP = high-pressure processing; TVC = total viable counts; ENB = Enterobacteriaceae; LAB = lactic acid bacteria; YM = yeasts and moulds; SEM = standard error of the mean. ^a,b,c,d,e^ Mean values with different superscript letters within a row were significantly (*p* < 0.05) different.

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
