# Peer review of "Impact of Thermal and High-Pressure Treatments on the Microbiological Quality and In Vitro Digestibility of Black Soldier Fly (Hermetia illucens) Larvae"

_animals, 2020, doi:10.3390/ani10040682_

Round 1

Reviewer 1 Report

Overall a nice paper. Kudos to the authors. There are some minor grammar issues which I have noted below. The authors need to review and add 4 papers (see comment #8 below) that look at the effect of processing on digestibility (2 in vivo and 2 in vitro) of insects and adjust the discussion accordingly. I also would caution the authors about using the conversion factor of 4.76 based on only a single study given the amount of amino acid analysis of black soldier fly larvae now available in the literature.

Page

Line

Comment

1

19

Change “Black Soldier Fly” to “The Black Soldier Fly”

1

23

Change “studied” to “study”

1

25

Change The high-pressure” to “High-pressure”

1

27

Change “which may be translate into” to “resulting in”

2

51

Change “the soybean and fish” to “soybean and fish”

2

54

Delete becoming

2

59

Change “steady” to “consistent”

2

62

Suggest some other refences to support the statement that digestibility is or is not affected by processing. Based on these papers you might wish to modify this sentence as well as parts of your discussion.

Ekpo K.E. 2011. Effect of processing on the protein quality of four popular insects consumed in Southern Nigeria. Archives of Applied Science Research 3:307-326.

Kinyuru J.N., Kenji G.M., Njoroge S.M.Muhoho S.N., Ayieko M. 2010. Effect of processing methods on the in vitro protein digestibility and vitamin content of edible winged termite (Macrotermes subhylanus) and grasshopper (Ruspolia differens). Food Bioprocess Technology 3:778-782.

Poelaert C., Despret X., Sindic M., Beckers Y., Francis F., Portetelle D., Soyeurt H., Théwis A., and Bindelle J. (2017). Cooking has variable effects on the fermentability in the large intestine of the fraction of meats, grain legumes, and insects that is resistant to digestion in the small intestine in an in vitro model of the pig’s gastrointestinal tract. Journal of Agricultural and Food Chemistry 65: 435–444.

Poelaert C., Francis F., Alabi T., Caparros Megido R., Crahay B., Bindelle J., and Beckers Y. (2018).   Protein value of two insects, subjected to various heat treatments, using growing rats and the protein digestibility-corrected amino acid score. J of Insects as Food and Feed 4:77-87.

3

85

Change “on” to “in”

3

91-93

I would reconsider using the conversion factor proposed by Janssen. There are a number of papers looking at amino acid content of black soldier fly larvae (or dried meals from larvae) that show results at odds with this conversion factor. A review of 30 different paper (including several of the ones referenced in this manuscript) looking at amino acids content of black soldier fly larvae and prepupae by this reviewer showed an average amino acid recovery (calculated as sum of the amino acids divided by crude protein calculated as N x 6.25) of 84.9% with a range from 66.5 to 99.2% (as a reference the data from this paper would calculate out to a recovery of 84%. This suggests that the very low conversion factor proposed by Janssen might be a result of the methods used to analyze their insects (note they only analyzed 3 species). There is plenty of data in the literature that suggest 4.76 is far too low for most species of insects.

6

168-169

If I sum protein+fat+ash I get a value of 79.65 and adding the chitin estimate of 6.6 that only adds up to 86.2% so what is the rest? You can’t use the ADF or NDF numbers as those fractions contain significant amounts of protein as evidenced by their amino acid content (see Finke (2007). Estimate of chitin in raw whole insects. Zoo Biology 26:105-115 and Finke (2013). Complete nutrient content of four species of feeder Insects. Zoo Biol. 32:27-36 which includes data for black soldier fly larvae).

Note if I use N x 6.25 for protein I get a protein of 44.8% so using this value rather than 34.1 I now get a sum of 96.8% which seems more reasonable.

10

170-172

This is confusing since the authors don’t express the amino acid data consistently. Note essential AA are expressed as g/100 g CP, lysine is expressed as g/100 g DM and methionine and tryptophan expressed as g/100 g units not specified but presumably are per 100 g dry matter.

10

259

Change “IVTDMD of the” to “IVTDMD of”

10

262-263

Might want to modify this slightly taking into account the refences of Kinyuru (2010) and Poelaert (2017) mentioned earlier in my comments.

11

277

Change ”seemed to be increased” to “seemed to increase”

Reviewer 2 Report

Dear Authors,

The manuscript titled “Impact of thermal and high-pressure treatments on the microbiological quality and in vitro digestibility of Black Soldier Fly (Hermetia illucens) larvae” reports the results of a study which evaluated the nutritional composition of BSFL reared on brewer’s by-product submitted to thermal and high-pressure processing treatments on the microbial levels and in vitro digestibility in both ruminants and monogastric models.

In my opinion, the manuscript fits well to the Animals aims and scope, reporting primary results for a new source of protein as is the BSFL. The aims of study, stated by authors as “(i) the evaluation of the nutritive value of the BSF larvae reared on brewer's by product (BBP) as a potential feed ingredient and (ii) the investigation of the effects of TP and HPP treatments on BSFL microbial levels and in vitro dry matter digestibility employing ruminant and monogastric models”, are very well defined an aligned with the title, the method and the results reported in the manuscript. The conclusions add valuable knowledge to the field when authors affirm that “the high-pressure processing showed no clear improvement in terms of decontamination capacity and digestibility in comparison to heating treatment, which may be translated into a less cost-effective process for large-scale production of Black Soldier Fly larvae”.

Not to mention that I don't have any additional comment, in the Line 23 … "The present studied showed ..." can be revised for "The present study showed ...".

Reviewer 3 Report

This paper evaluates the nutritional composition of black soldier fly (BSFL) reared on brewer's by-product (BBP) and the impact of thermal and high-pressure processing (HPP) treatments on the microbial levels and in vitro digestibility in both ruminant and monogastric models. The results showed that BBP-reared BSFL contained a high level of protein, amino acids, lauric acid and calcium, and high counts of total viable counts, Enterobacteriaceae, lactic acid bacteria and yeasts and moulds. Furthermore, heat-treated samples did not result in any significant changes (P>0.05) on any of the in vitro digestibility models, whereas HPP showed increased and decreased ruminal and monogastric digestibility, respectively.  They concluded that HPP did not seem to be a suitable, cost-effective method as an alternative to heat-processing for the large-scale treatment of BSFL. The paper presents a well-designed study and advances the scientific knowledge. The structure and methodology are well developed, and the results are presented and discussed appropriately. However, there is not any statistical test testing the hypothesis that the sample is homogeneous (homogeneity of variances in the data). Equal variances on the data should be investigated before ANOVA analysis. Arginine is a conditionally essential amino acid (Table 2) and tyrosine and cysteine are semi-essential amino acids.  The authors should discuss more their amino acid and fatty acid values and results and compare them with those reported in published papers with black soldier fly/defatted black soldier fly and different substrates. The authors should clarify the above.
